# Effects of Dietary Zearalenone Exposure on the Growth Performance, Small Intestine Disaccharidase, and Antioxidant Activities of Weaned Gilts

**DOI:** 10.3390/ani10112157

**Published:** 2020-11-19

**Authors:** Xinglin Liu, Chang Xu, Zaibin Yang, Weiren Yang, Libo Huang, Shujing Wang, Faxiao Liu, Mei Liu, Yuxi Wang, Shuzhen Jiang

**Affiliations:** 1Shandong Provincial Key Laboratory of Animal Biotechnology and Disease Control and Prevention, Department of Animal Sciences and Technology, Shandong Agricultural University, No. 61 Daizong Street, Tai’an 271018, China; liuxinglin1998@163.com (X.L.); yzb204@163.com (Z.Y.); wryang211@163.com (W.Y.); huanglibo123@126.com (L.H.); wsjhao@sdau.edu.cn (S.W.); 13505388008@163.com (F.L.); liumay@sdau.edu.cn (M.L.); 2State Key Laboratory of Animal Nutrition, College of Animal Science and Technology, China Agricultural University, No. 2. West Road Yuanming Yuan, Beijing 100193, China; xuchang_001@163.com; 3Agriculture and Agri-Food Canada, Lethbridge Research and Development Centre, Lethbridge, AB T1J 4B1, Canada; Yuxi.wang@agr.gc.ca

**Keywords:** zearalenone, weaned gilts, growth performance, disaccharidase, heat shock protein 70

## Abstract

**Simple Summary:**

This study was conducted to assess the effects of Zearalenone (ZEA) exposure on the growth performance, small intestine disaccharidase, and antioxidant activities of weaned gilts. Twenty weaned gilts were randomly divided into control and ZEA treatment (1.04 mg/kg) groups. The data showed that 1.04 mg/kg ZEA in gilt’s diet could reduce the activity of disaccharidase enzymes and induce oxidative stress in the small intestine. Therefore, ZEA may induce intestinal injury by oxidative stress, or induce oxidative stress through intestinal injury, thus reducing the effect of animals on nutrient absorption.

**Abstract:**

Zearalenone (ZEA) is a secondary metabolite with estrogenic effects produced by *Fusarium* fungi and mainly occurs as a contaminant of grains such as corn and wheat. ZEA, to which weaned gilts are extremely sensitive, is the main *Fusarium* toxin detected in corn–soybean meal diets. Our aim was to examine the effects of ZEA on the growth performance, intestinal disaccharidase activity, and anti-stress capacity of weaned gilts. Twenty 42-day-old healthy Duroc × Landrace × Large White weaned gilts (12.84 ± 0.26 kg) were randomly divided into control and treatment (diet containing 1.04 mg/kg ZEA) groups. The experiment included a 7-day pre-trial period followed by a 35-day test period, all gilts were euthanized and small intestinal samples were collected and subjected to immunohistochemical and western blot analyses. The results revealed that inclusion of 1.04 mg/kg ZEA in the diet significantly reduced the activities of lactase, sucrase, and maltase in the duodenum, jejunum, and ileum of gilts. Similarly, the activities of superoxide dismutase and glutathione peroxidase in the duodenum, jejunum, and ileum, and activities of catalase in the jejunum and ileum were reduced (*p* < 0.05). Conversely, the content of malondialdehyde in the duodenum, jejunum, and ileum, and the integrated optical density (IOD), IOD in single villi, and the mRNA and protein expression of heat shock protein 70 (Hsp70) were significantly increased (*p* < 0.05). The results of immunohistochemical analyses revealed that the positive reaction of Hsp70 in the duodenum, jejunum, and ileum of weaned gilts was enhanced in the ZEA treatment, compared with the control. The findings of this study indicate the inclusion of ZEA (1.04 mg/kg) in the diet of gilts reduced the activity of disaccharidase enzymes and induced oxidative stress in the small intestine, thereby indicating that ZEA would have the effect of reducing nutrient absorption in these animals.

## 1. Background

*Fusarium* mycotoxins are secondary metabolites produced by fungi in the genus *Fusarium*, which often occur as contaminants of corn and other grains that are used as a main source of feed for livestock and poultry [1]. Among these, zearalenone (ZEA) is the most common *Fusarium* toxin detected in corn–soybean meal diets that are used extensively for swine production, and has been reported to cause substantial economic losses to the pig industry every year [2]. Its primary toxic effects on animals include damage to the reproductive organs and a reduction in reproductive capacity [3,4], oxidative stress [5], endocrine disorders, cytotoxicity [6], and immunotoxicity [7]. ZEA can be converted to a range of different metabolites in animals, which affect the digestion and absorption of nutrients and reduce production performance, and swine appear to be particularly sensitive to this toxin.

Previous studies have concluded that the intestinal tract is the target organ of ZEA and that the toxin leads to pathological intestinal damage [8]. It has been reported that ZEA may disrupt the activity of antioxidant enzymes through oxidative stress, thereby reducing the growth performance of animals [3]. In addition, ZEA can also disrupt the activity of digestive enzymes, thus hindering the processes of digestion and absorption. Currently, however, there is little information available regarding the changes of disaccharidase activity in the small intestine of piglets attributable to ZEA, and the underlying mechanisms remain to be elucidated.

Previous toxicity studies have indicated that at concentrations of 2.5 and 40 μm, ZEA reduced the survival of jejunal epithelial cells in piglets [9]. Furthermore, a study found that ZEA had adverse effects on the development of early porcine blastocysts [10], which is consistent with the observations of Kouadio et al. [11], who demonstrated that ZEA can cause lipid peroxidation and lead to cell death. Heat shock protein 70 (Hsp70) is a ubiquitous member of the heat shock protein family found in most types of animal cells, and has been found to play range of different physiological roles, such as enhancing immune function and as an antioxidant [12]. Notably, it was found that Hsp70 may play a key role in defense against cellular oxidative stress induced by ZEA [13]. Different stress can damage the gastrointestinal membrane of piglets and induce the expression of Hsp70. Previous study has also shown that overexpression of Hsp70 can reduce the damage of jejunum mucosa of weaned piglets induced by lipopolysaccharide [14]. However, the Hsp70 expression and its location in the small intestine of piglet need to be further explored.

The European Commission has limited the ZEA concentration in piglet diets to 0.1 mg/kg [15], while in China, the maximum ZEA in piglet diets is 0.15 mg/kg [16]. However, a five-year investigation carried out by our team showed that the positive detection rate of ZEA was 69.15%, and the average value of ZEA was 0.97 mg/kg [17]. Given that contamination of cereal feed with ZEA is an extremely common phenomenon, it is highly desirable to characterize the effects of this toxin and identify the underlying mechanisms. Therefore, in this study, we sought to examine the effects of 1.04 mg/kg ZEA on the growth performance of weaned gilts, as well as disaccharidase activity, antioxidative activity, and Hsp70 expression in the small intestine of these animals, thereby providing a scientific basis for guidance on the production of healthy pigs.

## 2. Materials and Methods

### 2.1. ZEA-Contaminated Diet and Mycotoxin Determination

The purified ZEA was dissolved in ethyl acetate and sprayed on talcum powder. ZEA premix (1000 mg/kg) was prepared, and the mixture was put into the fume hood for 12 h. The purpose was to evaporate the residual ethyl acetate. The 10 mg/kg ZEA corn premix was obtained by diluting ZEA premix (1000 mg/kg) with corn flour without toxins. Based on the above premix, 1.0 mg/kg ZEA feed was prepared. Before the implementation, a covered container was prepared for storing the experimental feed. In this paper, the amount of ZEA was based on the research results of other researchers such as Jiang et al. [3], Chen et al. [18], and Dai et al. [4]. The nutritional components were analyzed at the beginning and after the experiment. Qingdao entry exit inspection and Quarantine Bureau was entrusted for toxin analysis. Levels of ZEA and aflatoxin (AFL) were quantified using liquid chromatography in conjunction with fluorescence detection, affinity column chromatography, and the external standard method, whereas high-performance liquid chromatography tandem mass spectrometry with fluorescence detection, affinity column chromatography, and the external standard method were used to quantify the levels of fumonisin (FUM) and deoxynivalenol (DON) [19,20]. The minimum detection concentration is 1.0 μg/kg for AFL, 0.01 mg/kg for ZEA, 0.05 mg/kg for DON, and 0.1 mg/kg for FUM.

### 2.2. Experimental Design, Animals, and Management

The use and prevention of this study met the requirements of “guidelines for the care and use of laboratory animals”, which was proposed by the national standards and ethics committee of the people’s Republic of China, and the approval number is GB/T 35892-2018. Twenty 42-day-old healthy gilts (Duroc × Landrace × Large White), with an average body weight of 12.84 ± 0.26 kg, were randomly divided into two treatment groups. Based on NRC (2012), piglets were fed with basal diet supplemented with 0 (control) or 1.0 (ZEA1.0) mg/kg for 35 days, with 0 and 1.04 mg/kg as the analyzed ZEA concentrations. No other toxins were found in the diet of the two groups. Table 1 shows the nutritional levels and composition of the basic diet. In the course of the experiment, the animals were raised in the Institute of Animal Nutrition, Shandong Agricultural University. When raising piglets, the environment was controlled. All piglets were placed in different metabolic cages (0.48 m^2^). Each cage had a feeder and a nipple water dispenser. In the first 7 days of the test, the ambient temperature was controlled at 30 °C, and after 7 days, the temperature was controlled at 26 °C to 28 °C, and 65% was relative humidity. When raising piglets, we should vaccinate them regularly, do a good job in routine management, and provide them with sufficient drinking water. The piglets were fed at 7:00 a.m., 1:00 p.m., and 7:00 p.m. respectively according to the feed intake during the pre-test period. A small amount of leftover feed in the trough was ensured at 7:00 a.m. on the following day, and the leftover feed was collected and weighed before refeeding.

### 2.3. Growth Performance Determination

Piglets were weighed individually before the morning feeding on two consecutive days at the beginning (40 and 41-day-old) and end (76 and 77-day-old) of the 35-day feeding period to determine the average daily gain (ADG). The average daily feed intake (ADFI) of different sows can be determined, and the feed efficiency can be calculated according to ADFI and ADG.

### 2.4. Small Intestine Sample Collection

At the end of the test, piglets were fasted for 12 h and euthanized by electrocution (head only). The abdomen was opened and intestinal samples were taken no more than 15 min. The distal duodenum, middle jejunum, and middle ileum (5–6 cm) were quickly washed with ice phosphate buffered saline (PBS, pH 7.4) and transferred to a 10 mL Eppendorf tube at −80 °C for the further analyses of disaccharidase and antioxidant enzyme activities, gene expression, and western blotting. A fourth sample (3–5 cm) from each intestinal region was rapidly fixed in Bouin’s solution for 24 to 48 h and dehydrated in a graded alcohol series for immunohistochemical analysis.

### 2.5. Disaccharidase Activity

The tissue samples were thawed, rinsed with ice deionized water, dried with filter paper, homogenized with 0.02 mmol/L Tris-HCl (pH 7.4) at 1:10 (mg/mL), and centrifuged (10,000× *g*) at 4 °C for 15 min. The supernatant was analyzed for protein content and enzyme activities. Disaccharidase activity within each intestinal sample was examined via a glucose oxidase–peroxidase coupled assay (with sucrose, maltose, and lactose used as substrates for sucrase, maltase, and lactase, respectively) using a commercial kit A082 (Nanjing Jiancheng Bioengineering Institute, Nanjing, China). Total intestinal protein content was determined using the Coomassie Brilliant Blue using a commercial kit A045-2 (Nanjing Jiancheng Bioengineering Institute). Enzyme-coupled assays were performed in triplicate to account for inter-assay variation.

### 2.6. Antioxidant Activity and Malondialdehyde Content

Intestinal samples were prepared as described in the previous section. The activity of catalase (CAT), glutathione peroxidase (GSH-Px), superoxide dismutase (SOD), and contents of malondialdehyde (MDA) in collected supernatants were determined using respective assay kits (A007, A005, A001, A001, and A003, Nanjing Jiancheng Bioengineering Institute). The activity of CAT was estimated based on the levels of H_2_O_2_ remaining after enzyme catalysis and determined according to the method described by Góth [21]. The activity of GSH-Px, which catalyzes glutathione oxidation from the reduced tripeptide glutathione (GSH), was determined using the method described by Maral et al. [22]. The activity of superoxide dismutase (SOD) was determined by the method of Ōyanagui [23]. Malondialdehyde (MDA) content was quantified using the thiobarbituric acid-reactive (TBA) method described by Placer et al. [24]. 

### 2.7. Immunohistochemistry

The small intestinal tissue fixed in Bouin’s solution were trimmed, dehydrated by gradient alcohol, transparented by xylene, and then embedded in paraffin. The wax blocks were continuously sliced with a thickness of 5 μm. Six consecutive slices treated with polylysine were selected for Hsp70 immunohistochemical staining. In order to obtain the antigen, the tissue sections were pretreated with sodium citrate buffer. The buffer concentration was 0.01 mol/L, pH was 6.0, and the treatment time was 20 min. In order to block the nonspecific binding, 10% normal goat serum (zsgb-bio, Beijing, China) was used to culture. In order to block the endogenous peroxidase activity, 10% hydrogen peroxide (H_2_O_2_) was used for 1.5 h. Then, the polyclonal antibody Hsp70 (1:50, bm0368, BIOSS, Beijing, China) was incubated with sections at 4 °C for 12 h. The samples were tested for immunohistochemical analysis using tissue staining SP Kit (spn-9001, zsgb-bio, Beijing, China); polink-2 plus Polymer horseradish peroxidase (HRP) rabbit or mouse primary antibody detection system (pv-9002, zsgb-bio, Beijing, China). The sections were soaked in diaminobenzidine tetrachloride, and the DAB Kit (pa110, Tiangen, Beijing, China) was used, and the time was no more than 3 min. The distribution of positive cells should be dehydrated and sealed, and observed by microscope.

### 2.8. Integrated Optical Density Measurement

Heat shock protein 70 (Hsp70) labeling was examined using a Nikon ELIPSE 80i microscope (Nikon, Tokyo, Japan). For each specimen, stained sections (10 stained sections per treatment) were randomly selected and photographed. The number of Hsp70 antigen and staining was detected and the image was analyzed. The software was Image Pro Plus 6.0 image analysis software (media controls, Silver Spring, MD, USA). These yielded values of the total cross-sectional integrated optical density (IOD) and single villus IOD (SIOD) of Hsp70 in the duodenum, jejunum, and ileum of weaned gilts [25] were used to compare the amounts of staining of samples obtained from control and ZEA-treated gilts.

### 2.9. Total RNA Extraction, cDNA Preparation, and Quantitative Real-Time Reverse Transcription-Polymerase Chain Reaction (qRT-PCR)

The small intestine sample was taken out from −80 °C and cut into 50–100 mg. The total RNA was extracted with Trizol Kit (Invitrogen, Carlsbad, CA, USA). The purity of RNA was detected at the absorbance ratio of 260/280 nm. The ds-11 ultraviolet spectrophotometer (denovix, Wilmington, DE, USA) was used. If the ratio was between 1.8 and 2.0, it showed that the RNA had high purity. The specific primers for Hsp70 gene were designed by searching the gene sequence of GenBank. The software was primer6.0, and the primer synthesis was undertaken by Bioengineering Co., Ltd (Shanghai, China). The reference gene transcript was *GAPDH* Gene. The target gene was normally expressed by using this method. The primer sequences and lengths of the amplified product are shown in Table 2. Reverse transcription was carried out immediately after the determination of mass and concentration, and was performed according to the instructions of a PrimerScript@ RT Master Mix Perfect Real Time kit (DDR036A: TaKaRa, Dalian, China). The reaction system (20 μL) was prepared according to the instructions of a SYBR ^®^Premix Ex TaqTM Tli RNaseH Plus quantitative fluorescence quantitative kit (RR420A; TaKaRa, Dalian, China). For each sample, target and internal reference genes were analyzed in triplicate. The internal reference gene and target gene were divided into three parts. ABI 7500 (Applied Biosystems, Foster City, CA, USA) was used for amplification under the following conditions: initial denaturation at 95 °C for 30 s, 95 °C for 43 cycles, lasting for 5 s, 60 °C for 34 s, 95 °C for 15 s, °C for 60 s. The fluorescence information was detected at 60 ℃, and the relative mRNA level was determined by 2^−△△CT^ method [26].

### 2.10. Western Blotting

Small intestine total protein was extracted according to the instructions of a commercial kit (Beyotime, Shanghai, China), and concentrations were determined using a BCA protein determination kit (Beyotime, Shanghai, China). Having established the concentration of each sample, the concentration of aliquots was adjusted to 60 μg, and these samples were electrophoretically separated on polyacrylamide gels, and subsequently transferred to solid-phase transfer membranes (Solarbio, Beijing, China). The membranes were incubated in 10% skimmed milk powder for 2 h, washed 3 times with Tris-buffered saline containing Tween (TBST) after incubation, and then incubated again with primary antibodies [polyclonal mouse anti-Hsp70, 1:300 (BIOSS); and monoclonal anti-actin, 1:1000 (Beyotime, Shanghai, China)], diluted with primary antibody dilution buffer (Beyotime, Shanghai, China), at 4 °C overnight. After washing with TBST, the membranes were incubated with anti-rabbit IgG and anti-mouse IgG antibodies (1:3000; Beyotime, Shanghai, China), and diluted with secondary antibody dilution buffer (Beyotime, Shanghai, China), at 37 °C for 2.5 h. Thereafter, the membranes were immersed in a high-sensitivity luminescence reagent (BeyoECL Plus; Beyotime, Shanghai, China), exposed to film using the FusionCapt Advance FX7 system (Beijing Oriental Science and Technology Development Co., Ltd., Beijing, China). The software for image analysis is IPP 6.0 (Media Cybernetics, Inc., Rockville, MD, USA).

### 2.11. Statistical Analysis

All data are expressed as the mean ± standard error (SE). To determine differences between groups, the data were evaluated to be normal distribution and variance homogeneity before *t*-test, and statistically analyzed using a two-sample pairwise *t*-test with SAS 9.2 statistical software (SAS Inst. Inc., Cary, NC, USA), with a *p* value < 0.05 being considered indicative of a significant difference.

## 3. Results

### 3.1. Growth Performance

No significant differences (*p* > 0.05) were observed with respect to the ADFI, ADG or feed efficiency (ADG/ADFI) of the control and ZEA-treated gilts after the 35-day feeding period (Table 3).

### 3.2. Disaccharidase Activity

The activities of lactase, sucrase, and maltase in the duodenum, jejunum, and ileum of gilts fed a diet containing 1.04 mg/kg ZEA were significantly lower (*p* < 0.05) than those in gilts fed the control diet (Table 4).

### 3.3. Antioxidant Activity and MDA Content

The activities of CAT in the jejunum and ileum, and those of SOD and GSH-Px in the duodenum, jejunum, and ileum of gilts fed a diet containing 1.04 mg/kg ZEA were significantly lower than those in the control group (*p* < 0.05), whereas the contents of MDA in the duodenum, jejunum, and ileum were significantly higher in ZEA-treated gilts than those in the control gilts (*p* < 0.05, Table 5). 

### 3.4. Localization of Hsp70

The effects of ZEA on the distribution of Hsp70 in the duodenum, jejunum, and ileum of gilts are shown in Figure 1, Figure 2 and Figure 3, respectively. Our observations revealed that Hsp70 immunoreactive substances were mainly detected in the mucosal epithelium of the duodenal villi, whereas no Hsp70 was detected in the small intestinal glands. Compared with the control group, gilts fed a diet containing 1.04 mg/kg ZEA showed significantly higher Hsp70 immunoreactivity in the lamina propria of the duodenum (Figure 1), and the distribution of Hsp70 immunoreactivity in the jejunum was mainly concentrated in the mucosal epithelium of jejunal villi and glandular epithelial cells of the small intestine (Figure 2). The distribution of Hsp70 immunoreactive substances in the ileum was similar to that in jejunum, and was localized to the intestinal and glandular epithelia of the small intestine. Furthermore, Hsp70 immunoreactivity in lymphoid nodules of ZEA group gilts was notably stronger compared with that detected in control group gilts (Figure 3). Similarly, myenteric neurons showed a strong Hsp70-positive reaction, which was stronger in the ZEA treatment group.

The analysis of total immunoreactive integrated optic density (IOD) and single villi SIOD of Hsp70 immunoreactivity in the duodenum, jejunum, and ileum of gilts revealed stronger reactions in the ZEA-treated gilts than the control group gilts (*p* < 0.05), and this enhancement was most evident in the jejunum, followed by the ileum and duodenum (Table 6).

### 3.5. Expression of Hsp70 at the mRNA and Protein Level

The mRNA and protein expression of Hsp70 in the small intestine of gilts is shown in Figure 4. Compared with the control group, gilts fed a diet containing 1.04 mg/kg ZEA showed increases in the relative expression of Hsp70 mRNA and protein in all segments of the small intestine (*p* < 0.05).

### 3.6. Discussion

Fungi within the genus *Fusarium* are probably the most prevalent toxin-producing fungi causing contamination of cereal grains grown worldwide [27]. Among the mycotoxins produced by different species of *Fusarium*, the most representative are DON, ZEA, and FUM, which are toxic to both livestock and humans [28]. In our previous study, although each raw material was carefully selected, we still detected different levels of DON and FUM in the control group samples [18], thereby highlighting the ubiquitous nature of *Fusarium* mycotoxin contamination and the importance of mycotoxin-related research.

### 3.7. Growth Performance

There is a certain dispute as to the effects of ZEA on animal production performance. The previous studies have indicated that diets containing ZEA have little or no effect on the ADFI and ADG of gilts [29,30], whereas it has been reported that the feed intake, ADG, and feed reward of sows decreased with an increase of the level of dietary ZEA (from 3.0 to 9.0 mg/kg) [31]. In contrast, low-dose ZEA (1 mg/kg) has been found to improve animal production performance, and thus ZEA is often used as a growth-promoting additive in North America [32]. The results showed that the growth performance of weaned piglets did not change when treated with 1.04 mg/kg ZEA, which was consistent with the research results of Oliver et al. [33]. The team found that growth performance did not change when weaned piglets were given 1.5 and 2 mg/kg ZEA. Some researchers took other animals as samples and found that feeding 0.2602 mg/kg ZEA diet would not affect the animal’s feed intake [34]. 

Some authors have proposed that the toxic effects of ZEA could be attributed to the accumulation of toxins to a certain concentration within the body [35], and it can thus be speculated that the effects of ZEA on growth performance may be dose- and species-specific. However, with regard to the inhibitory effects of ZEA on the digestion and absorption of intestinal nutrients in weaned gilts, further studies are necessary to determine the specific underlying mechanisms.

### 3.8. Disaccharidase Activity

In feed, polysaccharide and oligosaccharide can be absorbed only after degradation by disaccharidase. Therefore, in the process of carbohydrate digestion and absorption, we need to give enough attention to disaccharidase. Disaccharidases are secreted not from digestive glands but from epithelial cells lining the intestinal mucosa, which are of particular importance in the study of animal intestinal nutrition and are often considered indicators of intestinal damage. In this regard, a few previous studies have examined the effects of ZEA on the activity of disaccharidases in weaned gilts. By analyzing other research data, it can be clear that there is a relationship between disaccharidase activity and ZEA efficacy [36]. Some studies have shown that when the content of ZEA is 20 and 30 mg/kg, the digestive enzyme activity in jejunum of mice can be significantly inhibited [37]. Some researchers have studied ZEA and found that it can regulate TGF signaling pathway and Wnt/β-Catenin signaling pathway in HepG2 cells. Therefore, it can be considered that ZEA can change the mechanism of disaccharide secretion by proliferating intestinal cells [38]. On the basis of the present study, we speculate that 1.04 mg/kg ZEA in diet may be sufficient to inhibit the expression of disaccharidase-related genes and proteins, resulting in a reduction of disaccharidase activities in the intestinal tract of weaned gilts; however, the underlying mechanism awaits further elucidation.

### 3.9. Anti-Stress Ability

Antioxidant enzymes, such as CAT, GSH-Px, and SOD, comprise the major defense system designed to counter the detrimental effects of excessive reactive oxygen species (ROS) generation and thus cellular lipid peroxidation [39]. Because of the existence of these enzymes, oxygen free radicals in organisms can be produced and eliminated continuously, and then they can be maintained in a balanced state. Mycotoxins, including ZEA, can have the effect of disrupting this balance, resulting in oxidative damage. Malondialdehyde (MDA) is the final product of lipid peroxidation and is often used as an indicator to monitor lipid peroxidation and the body’ damage [40]. It has been concluded that the intestinal tract is the primary barrier with respect to ZEA metabolism [41]. In distal intestinal segments, ZEA metabolites gain access to body tissues via the systemic circulation and are re-activated. When weaned piglets were fed with 2.0 mg/kg diet, the level of MDA in the intestine was increased, and the levels of SOD and GSH-Px were decreased [3]. Other researchers have pointed out that ZEA (10 μM) can significantly change the expression levels of cat and GSH PX in intestine [42]. There are also research data show that when the content of ZEA is 6.25, 12.5, and 25 μM, the intestinal SOD, cat, and GSH-Px levels of piglets can be changed, and then the intestinal cells of piglets will be damaged [43].

In this study, we found that CAT levels in ileum and jejunum decreased in sows, and GSH-Px and SOD levels in ileum, jejunum, and duodenum decreased. However, studies conducted to date have tended to indicate that ZEA may induce oxidative stress via a number of different mechanisms. It has been confirmed that ZEA (0, 535, 1041, 1548, 2002, and 2507 μg/kg) can lead to intestinal oxidative damage, apoptosis, and tight junction disruption in fish [40], whereas other studies found that 1.0 mg/kg dietary ZEA can damage intestinal villous epithelial cells and destroy intestinal gland tissue, leading to oxidative stress [20], which is consistent with the findings of the present study. A study indicated that ZEA (6 and 8 μg/mL) induced cytochrome oxidoreductase-dependent oxidative stress in intestinal epithelial cells in gilts and the death of IPEC-J2 cells, following activation of the p38/MAPK signal pathway to regulate autophagy [44]. Furthermore, another study demonstrated that ZEA (100 μmoL) can induce the production of ERα signals and the destruction of signal cascade components such as NF-κB, ERK1/2, dcdx2, and HIF1α in the intestine of pigs [45]. However, the specific molecular mechanisms underlying the estrogenic activity of ZEA associated with intestinal toxicity need to be further verified.

### 3.10. Hsp70 Expression

As the most important member of the heat shock protein family, Hsp70 functions as a molecular chaperone and plays immune response-related roles, which are closely associated with the anti-stress responses of animals. Under normal circumstances, Hsp70 is generally not expressed, and it is only when the body is under stress, that there is a rapid increase in expression levels [46]. Hsp70 can inhibit apoptosis, and its overexpression is considered to be a biomarker of toxicity-related induction [13]. Most studies have established that ZEA can induce the expression of Hsp70. For example, compared with the control group, it can be clear that in the uterus of weaning sows, the 1.04 mg/kg ZEA arginine treatment sample has a higher level of Hsp70 mRNA [20].

With regard to the mechanism whereby ZEA induces the up-regulated expression of Hsp70, it is speculated that ZEA induces stress via a disruption of serum hormones and an upregulation of growth hormone receptor expression [20]. In addition, it has been established that apoptosis induced by ZEA (60, 90, and 120 μmoL) is mediated dose dependently via a caspase-3- and caspase-9-dependent mitochondrial pathway [47]. In the current study, we found that in weaned gilts fed a ZEA-containing diet, the expression of intestinal Hsp70 mRNA and protein was higher than that in the control group, and that the levels of immune-positive reactants were also significantly enhanced, indicating that large amounts of Hsp70 protein had been synthesized to counter the effects of ZEA toxin-induced stress. Combined with the observed changes in antioxidant enzyme activity, we can speculate that ZEA causes damage through oxidation, which can trigger the upregulated expression of Hsp70. These findings are consistent with view of Kukreja et al. [48], who proposed that the potential protective mechanism of Hsp70 may be associated antioxidative processes. The results of our immunohistochemical analyses revealed that Hsp70 immunoreactive substances were primarily detected in the epithelia of the jejunum and ileum, villi, and lymphoid nodules, which may be ascribed to the fact that small intestinal villous epithelium plays a major role in digestion and absorption. As the portion of the intestinal tract that absorbs most feed nutrients, the jejunum would tend to be most seriously damaged by ZEA, whereas lymph nodes as immune organs may be able better resist toxin invasion. However, the specific mechanisms whereby ZEA induces intestinal epithelial cell injury have yet to be elucidated.

## 4. Conclusions

Through this study, it can be clear that when the concentration meets the condition of 1.04 mg/kg, the expression of heat stress protein 70 and the content of malondialdehyde in duodenum, jejunum, and ileum of weaned piglets can be affected, and the intestinal tract of the body will be damaged and oxidative stress will occur. Therefore, it can be considered that ZEA can promote oxidative stress reaction. In order to further study the relationship between the two, more in vitro studies are needed to clarify the relationship between oxidative stress and intestinal injury in pigs.

## Figures and Tables

**Figure 1 animals-10-02157-f001:**
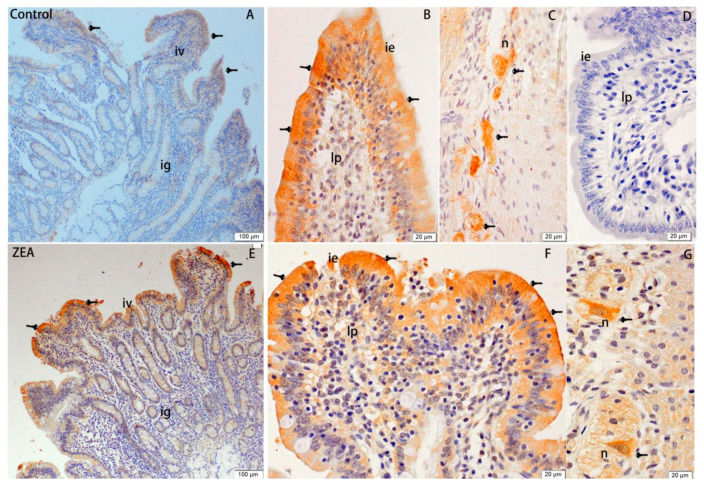
Effects of zearalenone (ZEA) on heat shock protein 70 (Hsp70) immuno reactivity distribution of duodenum in weaned gilts. (**A**–**C**) were control. (**E**–**G**) were ZEA treatment. (**D**) was the negative control. Scale bars were 100 μm for (**A**,**E**) and 20 μm for (**B**–**D**,**F**,**G**), respectively. The iv was intestinal villus, ig was intestinal glands, lp was lamina propria, and n was neuron.

**Figure 2 animals-10-02157-f002:**
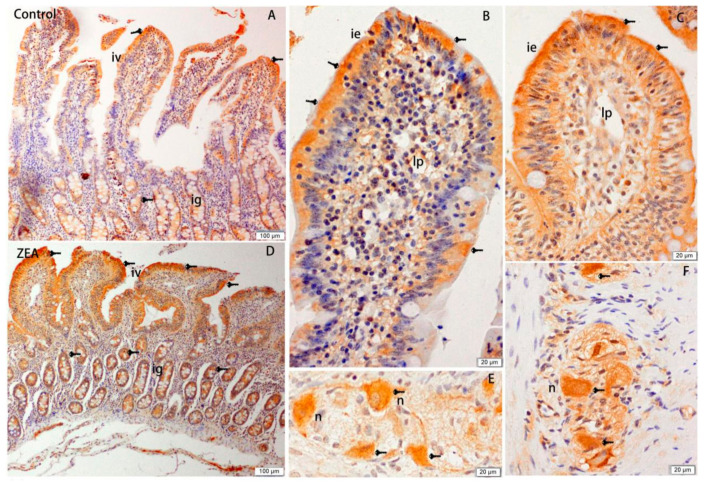
Effects of zearalenone (ZEA) on heat shock protein 70 (Hsp70) immuno reactivity distribution of jejunum in weaned gilts. (**A**,**B**,**E**) were control. (**C**,**D**,**F**) were ZEA treatment. Scale bars were 100 μm for (**A**,**D**), and 20 μm for (**B**,**C**,**E**,**F**), respectively. The iv was intestinal villus, ig was intestinal glands, lp was lamina propria, and n was neuron.

**Figure 3 animals-10-02157-f003:**
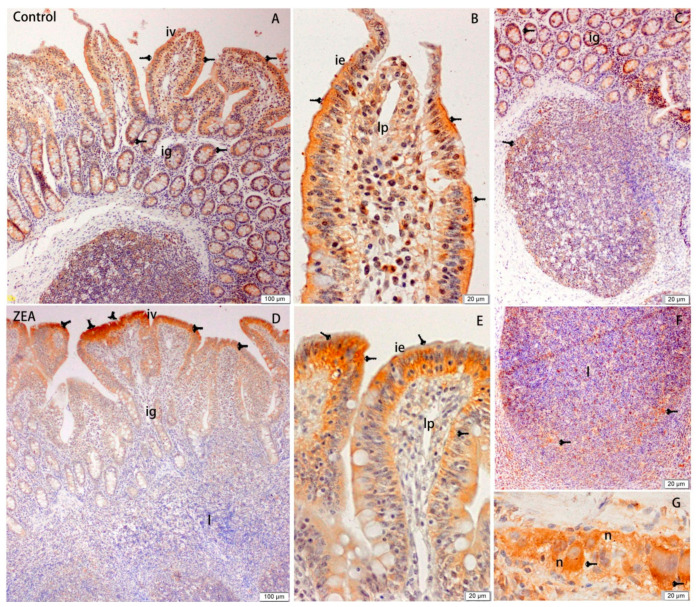
Effects of zearalenone (ZEA) on heat shock protein 70 (Hsp70) immuno reactivity distribution of ileum in weaned gilts. (**A**–**C**) were control. (**D**–**G**) were ZEA treatment. Scale bars were 100 μm for (**A**,**D**) and 20 μm for (**B**,**C**,**E**–**G**), respectively. The iv was intestinal villus, ig was intestinal glands, l was lymphoid nodule, lp was lamina propria, and m was musculari.

**Figure 4 animals-10-02157-f004:**
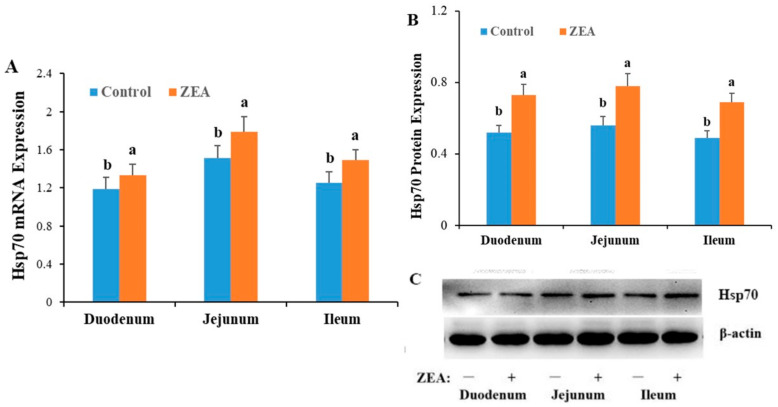
Effects of zearalenone (ZEA) on the mRNA and protein expressions of heat shock protein 70 (Hsp70) in duodenum, jejunum and ileum of weaned gilts. (**A**) was mRNA expression of Hsp70 in duodenum, jejunum, and ileum of weaned gilts. (**B**,**C**) were protein expression of Hsp70 in duodenum, jejunum, and ileum of weaned gilts. ^a, b^ Means within the same row differ significantly (*p* < 0.05).

**Table 1 animals-10-02157-t001:** Ingredients and compositions of the basal diet

Ingredients (%)	Content	Nutrients (%)	Analyzed Values
Corn	53.00	Metabolizable energy, MJ/kg	13.22
Sodium chloride	0.20	Tryptophan	0.25
Limestone, Pulverized	0.30	Threonine	0.90
Calcium phosphate	0.80	Sulfur amino acid	0.79
L-threonine	0.04	Methionine	0.46
DL-methionine	0.10	Lysine	1.36
L-Lysine HCl	0.30	Total phosphorus	0.73
Fish meal	5.50	Calcium	0.84
Soybean meal	24.76	Crude protein	19.40
Soybean oil	2.50		
Whey powder	6.50		
Wheat middling	5.00		
Premix ^1^	1.00		

^1^ Supplied per kg of diet: VA, 3,300 IU; D_3_, 330 IU; VE, 24 IU; K_3_, 0.75 mg; B_1_, 1.50 mg; B_2_, 5.25 mg; B_6_, 2.25 mg; B_12_, 0.02625 mg; pantothenic acid, 15.00 mg; niacin, 22.5 mg; biotin, 0.075 mg; folic acid, 0.45 mg; Mn (MnSO_4_·H_2_O), 6.00 mg; Fe (FeSO_4_·H_2_O), 150 mg; Zn (ZnSO_4_·H_2_O), 150 mg; Cu (CuSO_4_·5H_2_O), 9.00 mg; I (KIO_3_), 0.21 mg; Se (Na_2_SeO_3_), 0.45 mg.

**Table 2 animals-10-02157-t002:** Primers sequences of glyceraldehyde-3-phosphate dehydrogenase (GAPDH) and heat shock protein 70 (HSP70).

Target Gene	Primer Sequence (5′ to 3′)	Product Size	Accession No.
*GADPH*	F: ATGGTGAAGGTCGGAGTGAA	154	NM_001206359.1
	R: CGTGGGTGGAATCATACTGG		
*HSP70*	F: GAGGTGGAGAGGATGGTT	292	NM_001123127.1
	R: AGAGCCTGGAGAAGATGG		

**Table 3 animals-10-02157-t003:** Effects of zearalenone (ZEA) on the growth performance of post-weaning gilts.

Items	Control	ZEA
ADFI (g/d)	1070.39 ± 27.34	1007.07 ± 33.12
ADG (g/d)	548.78 ± 13.09	514.89 ± 20.78
FE	0.51 ± 0.02	0.51 ± 0.01

The results are presented by mean values ± standard deviation (n = 10). ADG, Average daily gain; ADFI, Average daily feed intake; FE, ADG g/ADFI g.

**Table 4 animals-10-02157-t004:** Effects of zearalenone (ZEA) on disaccharidase activity in the small intestine of post-weaning gilts.

Items (U/mg Protein)	Control	ZEA
Lactase		
Ileum	44.36 ± 1.55 ^a^	37.63 ± 1.90 ^b^
Jejunum	104.15 ± 12.87 ^a^	87.63 ± 14.03 ^b^
Duodenum	20.74 ± 0.897^a^	14.68 ± 0.65 ^b^
Sucrase		
Ileum	3.65 ± 5.55 ^a^	37.57 ± 4.09 ^b^
Jejunum	65.33 ± 2.99 ^a^	54.81 ± 5.18 ^b^
Duodenum	15.66 ± 1.43 ^a^	5.33 ± 0.51 ^b^
Maltase		
Ileum	62.57 ± 3.17 ^a^	52.70 ± 2.98 ^b^
Jejunum	159.88 ± 18.67 ^a^	116.20 ± 9.06 ^b^
Duodenum	63.33 ± 1.21 ^a^	52.50 ± 5.2 ^b^

The results are presented by mean values ± standard deviation (n = 10). ^a, b^ Means within the same row differ significantly (*p* < 0.05).

**Table 5 animals-10-02157-t005:** Effects of zearalenone (ZEA) on antioxidant enzyme activity and malondialdehyde content in the small intestine of post-weaning gilts.

Item	Control	ZEA
CAT (U/mg protein)		
Ileum	4.52 ± 0.14 ^a^	1.52 ± 0.13 ^b^
ejunum	7.35 ± 0.90 ^a^	3.95 ± 0.11 ^b^
Duodenum	28.68 ± 3.43	28.64 ± 2.19
SOD (U/mg protein)		
Ileum	43.24 ± 6.11 ^a^	36.01 ± 2.04 ^b^
Jejunum	76.57 ± 8.83 ^a^	42.95 ± 7.05 ^b^
Duodenum	152.24 ± 13.78 ^a^	125.08 ± 14.76 ^b^
GSH-Px (U/mg protein)		
Ileum	149.84 ± 15.44 ^a^	134.56 ± 13.70 ^b^
Jejunum	57.76 ± 3.98 ^a^	28.88 ± 5.12 ^b^
Duodenum	67.65 ± 9.02 ^a^	44.69 ± 8.45 ^b^
MDA (nmol/mg protein)		
Ileum	0.85 ± 0.07 ^b^	1.57 ± 0.23 ^a^
Jejunum	0.61 ± 0.02 ^b^	1.53 ± 0.11 ^a^
Duodenum	1.06 ± 0.06 ^b^	2.11 ± 0.02 ^a^

The results are presented by mean values ± standard deviation (n = 10). ^a, b^ Means within the same row differ significantly (*p* < 0.05). CAT, catalase; GSH-Px, glutathione peroxidase; SOD, superoxide dismutase; MDA, malondialdehyde.

**Table 6 animals-10-02157-t006:** Effects of zearalenone (ZEA) on immunoreactive intergrated optic density (IOD) and single villus’ IOD (SIOD) of heat shock protein 70 (Hsp70) in duodenum, jejunum, and ileum of weaned gilts (×10^3^).

Item	Control	ZEA
IOD in 10 × 10 visual field		
Duodenum	75.83 ± 12.44 ^b^	82.35 ± 16.14 ^a^
Jejunum	78.22 ± 9.78 ^b^	94.65 ± 10.23 ^a^
Ileum	77.17 ± 15.61 ^b^	93.57 ± 11.09 ^a^
SIOD in 10 × 40 visual field		
Duodenum	63.988 ± 5.11 ^b^	76.17 ± 7.93 ^a^
Jejunum	68.908 ± 10.32 ^b^	78.56 ± 8.17 ^a^
Ileum	65.929 ± 8.52 ^b^	74.44 ± 11.16 ^a^

The results are presented by mean values ± standard deviation (n = 10). ^a, b^ Means within the same row differ significantly (*p* < 0.05).

## Data Availability

The data appeared in this study are already publicly available in the literature.

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
