# Peer review of "Effects of Dietary Zearalenone Exposure on the Growth Performance, Small Intestine Disaccharidase, and Antioxidant Activities of Weaned Gilts"

_animals, 2020, doi:10.3390/ani10112157_

Round 1

Reviewer 1 Report

The paper titled “Effects of dietary zearalenone exposure on the growth performance, small intestine disaccharidase and antioxidant activities of weaned gilts” is well written but I have some comments and suggestions.

In the introduction section, the feeling of the reader is that several data on piglets are already available in literature and for this reason it could be very useful to underlying the relevance of this study compared to the others. Why do the authors prefer to use 1 mg/Kg of ZEA in this study? This aspect is clear in the discussion section but maybe it could be useful to have this information in the introduction. I think that could be interesting to know more detail on study plan in the introduction section.

In material and methods section, there are no mentions to ethical approval by appropriate ethic committee.

In the discussion section, sometimes it is very difficult to understand what authors want to underline. Maybe too many information at the same time are considered.

Figure 4 seems to me not at high definition.

Could you please explain this sentence? “The data appeared in this study are already publicly available in the literature”

Author Response

Reply to reviewer 1

----On behalf of my co-authors, we thank you very much for giving us an opportunity to revise our manuscript, we appreciate editor and reviewers very much for their positive and constructive comments and suggestions on our manuscript. We have checked the revised paper carefully, and we are looking forward to meeting the reviewers. If you have any questions, please don’t hesitate to let us know.

In the introduction section, the feeling of the reader is that several data on piglets are already available in literature and for this reason it could be very useful to underlying the relevance of this study compared to the others. Why do the authors prefer to use 1 mg/Kg of ZEA in this study? This aspect is clear in the discussion section but maybe it could be useful to have this information in the introduction. I think that could be interesting to know more detail on study plan in the introduction section.

----L67 (old) [L70-L73 (new)]: “The European Commission has limited the ZEA concentration in piglet diets to 0.1 mg/kg [14], while in China, the maximum ZEA in piglet diets is 0.15 mg/kg [15]. However, a 5-year investigation carried out by our team showed that the positive detection rate of ZEA was 69.15%, and the average value of ZEA was 0.97 mg/kg.” was added in the Background section.

In material and methods section, there are no mentions to ethical approval by appropriate ethic committee.

---- L92 (old) [L98-L100 (new)]: “All protocols used were consistent with the Guide for the Care and Use of Laboratory Animals and approved by the Committee on the Ethics (Approval Number: GB/T 35892-2018) of National standards of the People's Republic of China.” was added in the Material and Methods section.

In the discussion section, sometimes it is very difficult to understand what authors want to underline. Maybe too many information at the same time are considered.

----L317-L318 (old) [L317-L318 (new)]: “There is a certain degree of dispute as to the effects of ZEA on animal production performance.” was changed to “There is a certain dispute as to the effects of ZEA on animal production performance.”

----L318-L319 (old) [L317-L319 (new)]: “The findings of some previous studies have indicated that diets containing ZEA have little or no effect on the ADFI and ADG of gilts [26,27],” was changed to “The previous studies have indicated that diets containing ZEA have little or no effect on the ADFI and ADG of gilts [29,30],”

----L321-L322 (old) [L320-L321 (new)]: “In contrast, however, low-dose ZEA (1 mg/kg) has been found to improve animal production performance,” was changed to “In contrast, low-dose ZEA (1 mg/kg) has been found to improve animal production performance,”

----L323 (old) [L322-L323 (new)]: “The findings of the present study indicate that 1.04 mg/kg ZEA……” was changed to “The present study indicate that 1.04 mg/kg ZEA……”

----L325-L326 (old) [L324-L325 (new)]: “who found that ZEA at concentrations of between 1.5 and 2 mg/kg does not have a significant effect on the growth performance of weaned gilts.” was changed to “who found that 1.5 and 2 mg/kg ZEA does not have a significant effect on the growth performance of weaned gilts.”

----L327-L328 (old) [L326-L327 (new)]: “For example, a study found that low-dose ZEA (0.2602 mg/kg) diets had no significant effect on the egg production and feed intake of laying hens [31].” was changed to “For example, 0.2602 mg/kg ZEA diets had no significant effect on the egg production and feed intake of laying hens [34].”

----L335-L337 (old) [L334-L336 (new)]: “Oligosaccharides and polysaccharides in feed need to be degraded by the action of disaccharidase enzymes before they can be digested and absorbed by the body, and accordingly, the role of disaccharidases in carbohydrate digestion and absorption should not be underestimated.” was changed to “Oligosaccharides and polysaccharides in feed need to be degraded by disaccharidase before they can be absorbed. Therefore, the role of disaccharidases in carbohydrate digestion and absorption should not be underestimated.”

----L341-L342 (old) [L340-L3421(new)]: “One of the previous study, for example, has confirmed that the potency of ZEA is related to a reduction in disaccharidase activity [33]” was changed to “A previous study has confirmed that the potency of ZEA is related to the reduction of disaccharidase activity [36]”

----L343-L345 (old) [L341-L342 (new)]: “whereas in a study examining the single and combined toxic effects of different concentrations of ZEA (20 and 30 mg/kg) and DON (1.5 and 2.5 mg/kg) on mice, it was found that ZEA could reduce digestive enzyme activity in the jejunum of mice in the experimental groups [34].” was changed to “whereas it was found that ZEA (20 and 30 mg/kg) could reduce digestive enzyme activity in the jejunum of mice [37]”

----L346-L350 (old) [L342-L345 (new)]: “Furthermore, some researchers used an explant model to examine the changes in gene expression and protein richness in the jejunum of boars after acute ZEA poisoning, and accordingly found that ZEA can regulate the Wnt/β-catenin and TGF signal pathways, and confirmed that the associated changes can promote intestinal cell proliferation, which provides valuable insights for further studies on the mechanisms underlying the intestinal impairment associated with ZEA [35].” was changed to “In addition, researchers found that ZEA can regulate Wnt/β-Catenin and TGF signaling pathways in HepG2 cells, which provides a new idea for further study on the mechanism of ZEA changing the secretion of disaccharidase by promoting intestinal cell proliferation [38].”

----L350-L351 (old) [L345 (new)]: “On the basis of the findings of the present study, we speculate that 1.0 mg/kg ……” was changed to “On the basis of the present study, we speculate that 1.04 mg/kg ……”

----L358-L360 (old) [L352-L353 (new)]: “The activities of these enzymes provide an intracellular protective system that contributes to maintaining a dynamic balance between the production and scavenging of oxygen free radicals in organisms.” was changed to “The activities of these enzymes help to maintain the dynamic balance of oxygen free radical production and elimination in organism.”

----L362-L363 (old) [L355-L356 (new)]: “as an indicator molecule to monitor the level of lipid peroxidation and the degree of damage to the body [37]” was changed to “as an indicator to monitor lipid peroxidation and the body’ damage [40]”

----L365-L368 (old) [L358-L360 (new)]: “In a study in which weaned gilts were fed a diet containing 1 mg/kg ZEA for 28 days, significant reductions have been detected in the total antioxidant capacity of blood and reductions in the activities of GSH-Px and SOD, whereas there was a significant increase the content of MDA [39].” was changed to “The significant reductions of GSH-Px and SOD, and a significant increase of MDA were observed when weaned gilts were fed a ZEA (1 mg/kg) diet for 28 days [3].”

----L368-L369 (old) [L360-L361 (new)]: “Similarly, some studies found that dietary ZEA (10 μM) significantly altered……” was changed to “Similarly, some studies found that ZEA (10 μM) significantly altered……”

----L370-L373 (old) [L362-L363 (new)]: “whereas others demonstrated that ZEA (6.25, 12.5, and 25 μM) and its metabolites induces the production of intracellular ROS, and that antioxidant enzymes such as CAT, GSH-Px, and SOD contribute to reducing the cellular damage attributable to ZEA and its metabolites [41].” was changed to “whereas others demonstrated that antioxidant enzymes such as CAT, GSH-Px, and SOD could reduce ZEA (6.25, 12.5, and 25 μM) and its metabolites induced cell damage [43].”

----L374 (old) [L364 (new)]: “In the current study, we observed significant reductions in the activities of CAT…...” was changed to “In the current study, we observed significant reductions of CAT……”

----L387 (old) [L377 (new)]: “……ZEA associated with intestinal toxicity need to be further studied and verified.” was changed to “……ZEA associated with intestinal toxicity need to be further verified.”

----L397-L399 (old) [L385-L386 (new)]: “For example, an increase has been observed in the MDA content along with a rapid increasing in the expression of Hsp70 of uterus in weaned gilts fed a diet containing 1.0 mg/kg ZEA, thereby indicating a possible relationship between Hsp70 activity and oxidative stress [5]. Similarly, studies have detected a significantly higher relative expression of Hsp70 mRNA in the uterus of weaned gilts treated with 1.0 mg/kg ZEA compared with the control [17].” was changed to “For example, studies have detected a significantly higher relative expression of Hsp70 mRNA in the uterus of weaned gilts treated with 1.04 mg/kg ZEA compared with the control [20].”

Figure 4 seems to me not at high definition.

---- L301-L302 (old) [L48-L57 (new)]: Figure 4 is redrawn to improve the clarity.

Could you please explain this sentence? “The data appeared in this study are already publicly available in the literature”

---- I'm very sorry for the wrong choice when I submitted. The data presented in this study have not been published in the literature.

Reviewer 2 Report

Review of the manuscript No 986730 „Effects of dietary zearalenone exposure on the growth performance, small intestine disaccharidase and antioxidant activities of weaned gilts”

The work concerns an important issue - the quality of feeds and their nutritional safety in the human food chain. The problem of mycotoxin contamination of feeds is common around the world and, with maize being one of the most abundant feeds in pigs, zearalenone is an important toxin in pigs. The work is well prepared and has a detailed methodology. However, the manuscript needs minor revision.

Please, justify your ZEA dose selected, not just cite the literature (line 80). Especially in relation to the maximum dose allowed in feeds for pigs, which is actually lower than that chosen by the authors. The authors of the study cite other studies in which even lower doses of the ZEA had negative effects on the animal organism, why should it be different or worse at 1 mg/kg? A clear explanation of the chosen ZEA concentration should be included in the text.

In my opinion the information about the planned ZEA concentration (1mg/kg) and analyzed concentration (1.04mg/kg) in the M&M chapter is quite enough. It is not necessary to repeat this under each table. I suggest to standardize the description of ZEA concentration throughout the manuscript (especially under the tables): either the assumed amount of 1 mg/kg or the analytical amount 1.04 mg/kg.

In the presented experiment the ad libitum feeding system was used, it means that the access to feed is constant and it is available around the day and night. How did the authors technically carry out the pigs weighing procedure “before the morning feeding” and how did they carry out the control of daily amount of taken and left feed? Please, add the description.

Table 1: Control of the metabolizable energy coverage is used in the balancing of pig mixes. Please, provide the concentration of metabolizable energy of the mixture, not gross energy, in table 1.

Conclusions chapter must be changed. The conclusions in their current form are only a repetition of the results. Please, indicate what specific conclusion actually results from the conducted research and how it relates to the assumed experimental goal. It is also recommended to provide some practical conclusion for the pig feeding practice, if possible.

Author Response

Reply to reviewer 2

----On behalf of my co-authors, we thank you very much for giving us an opportunity to revise our manuscript, we appreciate editor and reviewers very much for their positive and constructive comments and suggestions on our manuscript. We have checked the revised paper carefully, and we are looking forward to meeting the reviewers. If you have any questions, please don’t hesitate to let us know.

Please, justify your ZEA dose selected, not just cite the literature (line 80). Especially in relation to the maximum dose allowed in feeds for pigs, which is actually lower than that chosen by the authors. The authors of the study cite other studies in which even lower doses of the ZEA had negative effects on the animal organism, why should it be different or worse at 1 mg/kg? A clear explanation of the chosen ZEA concentration should be included in the text.

----L67 (old) [L70-L73 (new)]: “The European Commission has limited the ZEA concentration in piglet diets to 0.1 mg/kg [14], while in China, the maximum ZEA in piglet diets is 0.15 mg/kg [15]. However, a 5-year investigation carried out by our team showed that the positive detection rate of ZEA was 69.15%, and the average value of ZEA was 0.97 mg/kg.” was added in the Background section.

In my opinion the information about the planned ZEA concentration (1mg/kg) and analyzed concentration (1.04mg/kg) in the M&M chapter is quite enough. It is not necessary to repeat this under each table. I suggest to standardize the description of ZEA concentration throughout the manuscript (especially under the tables): either the assumed amount of 1 mg/kg or the analytical amount 1.04 mg/kg.

---- L106-L107 (old), L225-L226 (old), L235-L236 (old), L247-L248 (old), L271-L272 (old), L279-L280 (old), L286-L287 (old), L294-L295 (old), L305-L306 (old): “1Treatments were basal diet supplemented with ZEA at the level of 0 and 1 mg/kg, with analyzed ZEA concentrations of 0 and 1.04 mg/kg, respectively.” below table 1, below table 3, below table 4, below table 5, below table 6, Figure 1, Figure 2, Figure 3 and Figure 4 was deleted.

---- ZEA concentration of 1.0 mg/kg throughout the manuscript was changed to “1.04 mg/kg”

In the presented experiment the ad libitum feeding system was used, it means that the access to feed is constant and it is available around the day and night. How did the authors technically carry out the pigs weighing procedure “before the morning feeding” and how did they carry out the control of daily amount of taken and left feed? Please, add the description.

---- [L113-L115 (new)]: “The piglets were fed at 7:00 a.m., 1:00 p.m. and 7:00 p.m. respectively according to the feed intake during the pre-test period. A small amount of leftover feed in the trough was ensured at 7:00 a.m. on the following day, and the leftover feed was collected and weighed before refeeding.” was added in the Material and Methods section.

---- L113-L114 (old) [L122-L123 (new)]: “Piglets were weighed before the morning feeding on two consecutive days at the beginning  and end of the 35-day feeding period to determine the average daily gain (ADG).” was changed to “Piglets were weighed individually before the morning feeding on two consecutive days at the beginning (40 and 41-day-old) and end (76 and 77-day-old) of the 35-day feeding period to determine the average daily gain (ADG).”

Table 1: Control of the metabolizable energy coverage is used in the balancing of pig mixes. Please, provide the concentration of metabolizable energy of the mixture, not gross energy, in table 1.

----L106 (old) [L105 (new)]: “Gross energy (17.12 MJ/kg)” was changed to “Metabolizable energy (13.22 MJ/kg)”.

Conclusions chapter must be changed. The conclusions in their current form are only a repetition of the results. Please, indicate what specific conclusion actually results from the conducted research and how it relates to the assumed experimental goal. It is also recommended to provide some practical conclusion for the pig feeding practice, if possible.

----L420-L424 (old) [L407-L413 (new)]: “In this study, we found that, at a concentration of 1.04 mg/kg, dietary ZEA induced intestinal injury and oxidative stress in weaned gilts, and significantly reduced the activities of disaccharidase (lactase, sucrase and maltase) and antioxidant (catalase, glutathione peroxidase, superoxide dismutase) enzymes in the duodenum, jejunum, and ileum of weaned gilts, and increased the levels of malondialdehyde and expression of heat stress protein 70 at both mRNA and protein levels.” was changed to “In this study, we found that, at a concentration of 1.04 mg/kg, dietary ZEA induced intestinal injury and oxidative stress by decreasing the enzymes activities (disaccharidase and antioxidant enzymes) and increasing the malondialdehyde and heat stress protein 70 expression in the duodenum, jejunum, and ileum of weaned gilts. It is speculated that the mechanism of ZEA induced intestinal injury may be related to the oxidative stress. Further in vitro studies are needed to determine the relationship between intestinal injury and oxidative stress in gilts challenged by ZEA.”

Reviewer 3 Report

I have reviewed the manuscript and I think it is well written and structured. The topic is of avarage interest and, upond some modifications, suitable for publication.

The introduction may be implemented with some more data regarding similar trials in piglets. The section introducing HSP is quite poor since a lot of work has been done in the swine species.

The experimental desing is appropriate and well described but I think that some methods sections may be implemented. I think it would be important to state the sex of the animals, even if young and not sexually mature. The sexes ratios in the 2 experimental groups should be reported. Moreover, how were the animals euthanized? This may affect the sampling procedures, therefore it needs to be discussed.

Also regarding animals, there is no mention to an ethical approval. Please add it either in the methods or at the end of the manuscript befere the references.

The statistical analysis needs to be implemented. T tests are appropriate only if data is both normally distributed and homoscedastic. Did the authors perform such evaluations?

Please check for tables formatting and image quality, as they are often poor.

Author Response

Reply to reviewer 3

----On behalf of my co-authors, we thank you very much for giving us an opportunity to revise our manuscript, we appreciate editor and reviewers very much for their positive and constructive comments and suggestions on our manuscript. We have checked the revised paper carefully, and we are looking forward to meeting the reviewers. If you have any questions, please don’t hesitate to let us know.

The introduction may be implemented with some more data regarding similar trials in piglets. The section introducing HSP is quite poor since a lot of work has been done in the swine species.

---- [L66-L69 (new)]: “Different stress can damage the gastrointestinal membrane of piglets and induce the expression of Hsp70. Previous study has also shown that overexpression of Hsp70 can reduce the damage of jejunum mucosa of weaned piglets induced by lipopolysaccharide [14]. However, the Hsp70 expression and its location in the small intestine of piglet needs to be further explored.” was added in the Material and Methods section.

The experimental desing is appropriate and well described but I think that some methods sections may be implemented. I think it would be important to state the sex of the animals, even if young and not sexually mature. The sexes ratios in the 2 experimental groups should be reported. Moreover, how were the animals euthanized? This may affect the sampling procedures, therefore it needs to be discussed.

----L92-L93 (old) [L100-L101 (new)]: “Twenty 42-day-old healthy gilts (Duroc × Landrace × Large White), with an average body weight of 12.84 ± 0.26 kg, were randomly divided into two treatment groups.” Here gilt is female piglet. The female piglets were selected in this experiment.

----L118 (old) [L128-L129 (new)]: “At the end of the experiment, gilts were fasted for 12 h and then euthanized.” was changed to “At the end of the experiment, gilts were fasted for 12 h and then euthanized by electrocution (head only).”

Also regarding animals, there is no mention to an ethical approval. Please add it either in the methods or at the end of the manuscript befere the references.

---- L92 (old) [L98-L100 (new)]: “All protocols used were consistent with the Guide for the Care and Use of Laboratory Animals and approved by the Committee on the Ethics (Approval Number: GB/T 35892-2018) of National standards of the People's Republic of China.” was added in the Material and Methods section.

The statistical analysis needs to be implemented. T tests are appropriate only if data is both normally distributed and homoscedastic. Did the authors perform such evaluations?

----L216-L217 (old) [L226-L227 (new)]: “To determine differences between groups, the data were statistically analyzed using……” was changed to “To determine differences between groups, the data were evaluated to be normal distribution and variance homogeneity before t-test, and statistically analyzed using……”

Please check for tables formatting and image quality, as they are often poor.

---- The processing of table format and image quality is completed again.

Round 2

Reviewer 1 Report

to my opinion the revised paper is fine

Author Response

Dear Ms. Lico Liao,

Thank you very much for giving us the opportunity to revise the manuscript again. We have reduced the repetition rate of manuscripts to less than 30%. We re submit the revised manuscript in the manuscript submission system, expecting the manuscript to meet the requirements of the magazine. If you have any questions, please let us know.

Kind regards,

Shuzhen Jiang

Reviewer 3 Report

The authors have addressed my concerns.

Author Response

(The authors gave the same response as above.)
